# HIV-1 *Trans* Infection via TNTs Is Impeded by Targeting C5aR

**DOI:** 10.3390/biom12020313

**Published:** 2022-02-15

**Authors:** Giulia Bertacchi, Wilfried Posch, Doris Wilflingseder

**Affiliations:** Institute of Hygiene and Medical Microbiology, Medical University of Innsbruck, 6020 Innsbruck, Austria; giulia.bertacchi@i-med.ac.at (G.B.); wilfried.posch@i-med.ac.at (W.P.)

**Keywords:** HIV-1, complement, tunneling nanotubes (TNTs), *trans* infection

## Abstract

Nonadjacent immune cells communicate through a complex network of tunneling nanotubes (TNTs). TNTs can be hijacked by HIV-1, allowing it to spread between connected cells. Dendritic cells (DCs) are among the first cells to encounter HIV-1 at mucosal sites, but they are usually efficiently infected only at low levels. However, HIV-1 was demonstrated to productively infect DCs when the virus was complement-opsonized (HIV-C). Such HIV-C-exposed DCs mediated an improved antiviral and T-cell stimulatory capacity. The role of TNTs in combination with complement in enhancing DC infection with HIV-C remains to be addressed. To this aim, we evaluated TNT formation on the surface of DCs or DC/CD4^+^ T-cell co-cultures incubated with non- or complement-opsonized HIV-1 (HIV, HIV-C) and the role of TNTs or locally produced complement in the infection process using either two different TNT or anaphylatoxin receptor antagonists. We found that HIV-C significantly increased the formation of TNTs between DCs or DC/CD4^+^ T-cell co-cultures compared to HIV-exposed DCs or co-cultures. While augmented TNT formation in DCs promoted productive infection, as was previously observed, a significant reduction in productive infection was observed in DC/CD4^+^ T-cell co-cultures, indicating antiviral activity in this setting. As expected, TNT inhibitors significantly decreased infection of HIV-C-loaded-DCs as well as HIV- and HIV-C-infected-DC/CD4^+^ T-cell co-cultures. Moreover, antagonizing C5aR significantly inhibited TNT formation in DCs as well as DC/CD4^+^ T-cell co-cultures and lowered the already decreased productive infection in co-cultures. Thus, local complement mobilization via DC stimulation of complement receptors plays a pivotal role in TNT formation, and our findings herein might offer an exciting opportunity for novel therapeutic approaches to inhibit *trans* infection via C5aR targeting.

## 1. Introduction

With 1.5 million new infections in 2020 [1], the HIV-1 pandemic is far from over, with the genetic variability and the global diversity of the virus challenging the development of a curative treatment or vaccine [2].

Although the major targets of HIV-1 infection are CD4^+^ T cells, dendritic cells (DCs) play a crucial role in HIV-1 transmission and the shaping of adaptive immunity [3,4]. Indeed, DCs are among the first cells to encounter HIV-1 at the mucosal surface, where they act as a surveillance system, taking up antigens at the site of the infection [5]. Upon contact with the invading pathogen, DCs mature into potent antigen-presenting cells (APCs) and present the acquired antigen to naive CD4^+^ T cells either in mucosal tissue or after migrating to lymph nodes [6,7,8,9].

Despite DCs being a vehicle for trans infection, they only show low-level productive infection with HIV-1 due to a number of restriction factors [10,11]. Among the most potent restriction mechanisms of HIV-1 infection in DCs is SAM domain HD domain-containing protein 1 (SAMHD1) [12,13]. SAMHD1 blocks HIV-1 at the post-entry step by reducing the intracellular dNTP pool, hence depriving the virus of the required ‘building blocks’ for its replication [14,15,16].

Interestingly, complement-opsonized HIV-1 (HIV-C) efficiently infects DCs by evading SAMHD1 restriction due to phosphorylation of the T592 residue of the SAM domain, resulting in improved antiviral immune responses [17,18,19,20,21,22,23]. Moreover, DCs exposed to HIV-C display enhanced maturation and costimulatory capacity [17], as well as the production of type I interferons, compared to DCs exposed to nonopsonized HIV-1, which do not produce type I or III interferons [18,19,24]. On the contrary, another study recently illustrated that complement opsonization supports HIV establishment in colorectal mucosa, therefore creating an environment with higher levels of infection compared to the nonopsonized virus [25].

Complement opsonization of HIV-1 is a relevant process in the outcome of viral infection. In fact, the initial host-HIV-1 interaction includes the activation of the complement system via all three signaling cascade pathways [26], which also happens in semen [27]. However, HIV-1 has evolved mechanisms to elicit complement-mediated destruction by incorporating regulators of complement activation during the budding process and additionally binding the fluid-phase factor H [28,29]. Thus, only low levels of HIV-1 particles are destroyed by complement-mediated lysis [29].

In addition, as the viral envelope possesses a complement-binding domain, HIV-C spontaneously accumulates as early as during transmission and the acute phase of infection [18,30,31,32,33]. Complement deposition on its surface protects the virus from complement-mediated neutralization [34] and increases HIV-1 infectivity through the interaction with complement receptors [22,35].

In 2004, Gerdes and colleagues described a novel route of cell-to-cell communication: long cytoplasmic bridges, mainly made of F-actin, named tunneling nanotubes (TNTs) [36]. TNTs allow the formation of a complex cellular network [37], which, in physiological conditions, facilitates fast and specific responses between different cells through the exchange of organelles, vehicles and the coordination of signals [38]. These structures can either be open-ended, supporting the continuity of the cytoplasm, or closed-ended, allowing only molecular exchange between nonadjacent cells. Immune cells, including B cells, macrophages, dendritic cells, T cells and NK cells, can all form TNTs (as reviewed in [39]), which have particularly been observed between immune cells in lymph nodes [40], as well as between DCs resident in the gut, where they cross the epithelial layer into the gut lumen to access bacteria [41].

Besides their physiological role, TNTs can be hijacked by invading pathogens, which then spread between cells protected by the extracellular environment. In this context, TNTs have previously been demonstrated to favor HIV-1 spread between macrophages and T cells and also between T cells themselves [42,43,44,45,46,47,48]. Indeed, it has been calculated that transmission of HIV between T cells through TNTs is 100- to 1000-fold higher than through classical cell-free infection [49]. The influence of actin remodeling on HIV infection has previously been demonstrated in CD4^+^ T cells, where dynamin inhibition has been linked to infection of impaired cells [50]. In DCs, budding HIV and immature HIV particles have been observed moving inside long actin-rich filopodia, inconsistent with the TNT structure, which converge to form viral synapses with CD4^+^ T cells [51]. Additionally, DCs have recently been described to display a higher number of TNTs when in co-culture with CD4^+^ T cells and in the presence of IFN-γ. HIV-like particles were found inside these structures, which spanned from one DC to another [52].

To date, the role of complement in the formation of TNTs has not yet been addressed. To this aim, here, we compared the number of TNTs displayed on the surfaces of DCs infected with HIV or HIV-C and collected viral supernatants in parallel as a measure of infection. A similar experiment was performed in which HIV/HIV-C DCs were incubated with either cytarabine (AraC) or TNTi, which are known to inhibit TNT formation by respectively interfering with the NF-kB pathway [53] and M-Sec-dependent TNT formation [43]. Lastly, the role of local complement activation induced in HIV- or HIV-C-exposed DCs or DC/T-cell co-cultures and their role in TNT formation as well as viral spread were addressed by incubating infected cells with inhibitors of either C3a or C5a receptors (C3aR and C5aR).

## 2. Materials and Methods

### 2.1. Ethics Statement

All participating blood donors provided written informed consent to the Central Institute for Blood Transfusion and Immunological Department, Innsbruck, Austria. The use of anonymized leftover specimens for research on host/pathogen interactions was approved by the Ethics Committee of the Medical University of Innsbruck (ECS: 1166/2018).

### 2.2. Generation of Human Monocyte-Derived DCs and CD4^+^ T Cells

To isolate monocytes, blood was received by the Central Institute for Blood Transfusion and Immunological Department, Innsbruck, Austria. Briefly, peripheral blood mononuclear cells (PBMCs) were isolated from the blood of healthy donors [54] by density gradient centrifugation using a Ficoll Paque Premium (GE Healthcare, Chicago, IL, USA) gradient. After washing, CD14^+^ monocytes were isolated from PBMCs using anti-human CD14 Magnetic Beads (BD); the purity of the isolated cells was assessed by FACS and was at least 98%. Monocytes were stimulated by adding IL-4 (200 U/mL) and GM-CSF (300 U/mL) for 5 days to generate iDCs, which were then harvested and used for all further experiments.

CD4^+^ T cells were isolated from PBLs using a human naive CD4^+^ T-cell enrichment set (BD). The purity of the isolated cells was assessed by fluorescence-activated cell sorting (FACS) and was at least 98%. CD4^+^ cells were stimulated with 200 U/mL IL-4 for one day before being used in the following experiments.

### 2.3. Multicolor Fluorescence-Activated Cell Sorting (FACS) Analyses

To assess their purity, multicolor fluorescence-activated cell sorting (FACS) analyses were performed on monocytes immediately after their isolation using antibodies against the following lineage markers: CD3 for T lymphocytes, CD14 for monocytes and CD19 for B lymphocytes. FACS analyses of monocyte-derived DCs were performed on day 5 to measure the expression levels of CD11c, CD83 and DC-SIGN.

FACS analyses of isolated CD4^+^ T cells were performed to measure the expression levels of CD3, CD4 and CD8.

FACS analyses were performed as previously described [55] on a FACSVerse flow cytometer (BD Biosciences, Franklin Lakes, NJ, USA). Data were analyzed using FACSDiva Software version 8 (BD Biosciences) [56].

### 2.4. Plasmids

The infectious R5-tropic HIV-1 proviral clone R9Bal was used for maturation, binding/internalization and DC infection studies.

Confocal microscopic analyses and HC/HT imaging analyses were performed by using chimeric R9Bal/mCherry and R9Bal/GFP virus preparations originating from R9Bal and vpr/mCherry or /GFP expression plasmids.

### 2.5. Virus Production

HIV-1 proviral clones were produced by transfecting HEK293T cells using the CaCl2 method [17]. Viral supernatants were collected on several days postinfection (dpi), cleared by filtration through 0.22 μm pore-size filters and concentrated by ultracentrifugation at 20,000 rpm for 90 min at 4 °C. The virus pellet was resuspended in RPMI1640 without supplements and stored in small aliquots at −80 °C to avoid multiple thawing. One aliquot was taken to determine the virus concentration by p24 ELISA [57].

### 2.6. Virus Opsonization

To mimic opsonization in vitro, purified HIV-1 and VLP stocks were incubated for 1 h at 37 °C with normal human serum (NHS) in a 1:10 dilution. As a negative control, the virus was incubated under the same conditions in culture medium. After opsonization, the virus was thoroughly washed to remove unbound components, pelleted by ultracentrifugation (20,000 rpm/90 min/4 °C) and resuspended in culture medium without supplements, and virus concentrations were determined using p24 ELISA. The opsonization pattern was analyzed using a virus capture assay (VCA), as described below.

### 2.7. p24 ELISA

To analyze supernatants for HIV p24 capsid protein, after virus production and following an infection experiment, a p24 sandwich ELISA was performed. Briefly, the ELISA plate was coated using a mouse MAb against HIV-1 p24 Ag (200 ng/well). For infection experiments, lysed samples (diluted 1:1 with 2% Igepal) were added for 1 h at room temperature. Bound HIV-1 p24 Ag was detected with a second biotinylated anti-p24 MAb followed by streptavidin–β-galactosidase conjugate. The color reaction was developed with the resorufin-β-d-galactopyranoside substrate (Sigma-Aldrich, Saint Louis, MI, USA), and the optical density was measured on an ELISA microplate reader at 550 nm. The amount of p24 was calculated from a standard curve by using recombinant HIV-1 p24 Ag. All reagents for p24 ELISA were kindly provided by Polymun Scientific, Klosterneuburg, Austria.

### 2.8. Virus Capture Assay

The opsonization pattern was determined by virus capture assay (VCA) as previously described by our team [20] using anti-human C3c (recognizing C3b, iC3b), C3d, IgG or mouse IgG Abs as controls for background binding. The coated VCA plates were incubated overnight with the differentially opsonized virus preparations (2.5 ng p24/well) at 4 °C and washed 5 times, and the bound virus was lysed and transferred to a precoated p24 ELISA plate.

### 2.9. Infection of Cells

In the first experimental setting, DCs were differentially matured (iDCs1:1000 iU/mL IFNγ, mDCs1: 1000 iU/mL IFNγ, 250 ng/mL LPS, IDCs2: 10^−6^ mol PGE2 mDCs: 10^−6^ mol PGE: 250 ng/mL LPS) for 1 day, co-cultured with CD4^+^ T cells and infected in duplicates using differentially opsonized HIV-1.

Briefly, in a 96-well U-bottom plate (Greiner Bio-One, Kremsmünster, Austria), 10^5^ cells in 100 µL/well (in a ratio of 2 CD4^+^ T cells/1 DC in co-cultures) were infected with 25 ng p24/mL of HIV/HIV-C or left uninfected. To confirm productive infection with HIV-1, we thoroughly washed the cells after 24 h and cultured cells at 37 °C/5% CO_2_. Using ELISA, we measured the p24 concentrations in the supernatants after spinning down the plate to pellet cells at 7 dpi.

In the second infection experiment, iDCs, CD4^+^ or DC/CD4^+^ co-cultures were seeded in a 96-well U-bottom plate (Greiner Bio-One) (10^5^ cells in 100 µL/well, in a ratio of 2 CD4^+^ T cells/1 DC in co-cultures) and infected with 25 ng p24/mL of HIV/ HIV-C or left uninfected. Each condition was then treated for 24 h with TNT inhibitors: cytarabine (1 µM, Sigma-Aldrich) or TNTi (20 µM, Pharmeks, Moscow, Russia). After 24 h, cells were thoroughly washed, and supernatants were collected as above at 7 days postinfection. Lastly, the infection experiment setting was repeated by preincubating cells with anaphylatoxin inhibitors (C3aR, C5aRI and C5aRII), both obtained from Sigma-Aldrich: C3aRi (SB 290157) and C5aRi (mix of W-54011 and DF2593, 1:1). Cells were treated for 2 h with 1 μM C3aR or C5aR antagonists before infection. After 3 h, cells were thoroughly washed and incubated with 0.1 µM anaphylatoxin receptor antagonists for the remaining days before supernatant retrieval.

### 2.10. Confocal Analysis

To quantify intracellular HIV/HIV-C and C3 induction by the differentially opsonized virus by confocal microscopy and to count TNTs, 50,000 cells/well were seeded in CellCarrier Ultra plates (Perkin Elmer, Waltham, MA, USA), following the three experimental protocols above, and infected with differentially opsonized and fluorescently labeled (GFP or mCherry) HIV-1. After infection, cells were fixed with 4% paraformaldehyde. Intracellular staining was performed using 1× Intracellular Staining Permeabilization Wash Buffer (10×; BioLegend, San Diego, CA, USA).

The following staining protocols were followed for each of the previously described experiments (see “Infection of Cells”). First and second experimental settings: Hoechst 33342 (Cell Signaling Technologies, Danvers, MA, USA), phalloidin-Alexa 647 (Cell Signaling Technologies and CD11c-PE (BD Biosciences) for 1 h at RT, then thoroughly washed in D-PBS.

Third experimental setting: Hoechst 33342 (Cell Signaling Technologies), phalloidin-Alexa 647 (Cell Signaling Technologies) and complement C3 (C3-FITC, Agilent Technologies, Santa Clara, CA, USA) for 1 h at RT, then thoroughly washed in D-PBS.

Image analysis was performed in a high-throughput manner using the Operetta CLS system (PerkinElmer). Spot analyses, nanotube evaluation and absolute quantification for C3-, HIV/HIV-C-, or C3-/HIV/HIV-C-containing cells were conducted using the Harmony software version 4.8 (Perkin Elmer) [58]. For quantification, at least 1000 cells per condition were analyzed.

In order to be counted as a TNT, no less than one membranous structure must be present that connects two cells, is at least partially nonadherent to the substratum and has a minimum length of 8 µm. Cells with no TNTs must be within one cell body length of another cell without touching any other cell to be classified as negative [59].

### 2.11. Statistical Analysis

Differences were analyzed by using GraphPad Prism software version 9 (GraphPad Software Inc., San Diego, CA, USA) [60] and one-way ANOVA with *Bonferroni post*-test for multiple comparisons or unpaired Student’s *t*-test depending on the analyses performed.

## 3. Results

### 3.1. HIV-C Enhances TNT Formation in DCs and DC/CD4^+^ T-Cell Co-Cultures

In the first step, we assessed the role of complement opsonization of HIV-1 in TNT formation. For this, immature (iDCs) and mature DCs (mDCs) were exposed to HIV or HIV-C and co-cultured with autologous naïve CD4^+^ T cells. The number of TNTs displayed on DCs was counted 7 days postinfection (dpi), the same day when productive infection of co-cultures was analyzed, since TNT formation in DCs reaches its maximum at 7 dpi compared to earlier time points (Appendix A). As a negative control for TNT formation, DCs were subjected to the chronic inflammatory mediator PGE_2_ at the early stages of maturation, which is known to generate DCs that fail to reticulate when cultured with CD4^+^ T cells [52]. Conversely, IFNγ, together with CD40L on the surface of CD4^+^ T cells, stimulates TNT formation in DCs [52], and thus, we used this condition as a positive control for DC reticulation. TNTs appear as straight membranous protrusions rich in F-actin that hover above the substratum and connect two distant cells, which can be assessed using high-content screening and digital phase contrast (DPC). HIV infection enhanced TNT formation in both iDCs and mDCs (Figure 1a), in line with previous observations in macrophages [42,43] and U87 cells [44].

Interestingly, HIV-C-loaded iDCs as well as mDCs displayed a significantly higher number of TNTs on their surface when co-cultured with CD4^+^ T cells (Figure 1a,b) which is particularly impressive when the inevitable loss of some of these structures during chemical fixation is considered [61]. Not only in DC/CD4^+^ T-cell co-cultures (Figure 1a,b) but also in cultures of DCs alone, HIV-C enhanced TNT formation, but at significantly lower levels compared to the DC/CD4^+^ co-culture (Figure 1c,d), highlighting the importance of CD4^+^ T cells for DC reticulation [52]. From qualitative observations, TNTs seem to especially form between heavily infected cells (Figure 1e). Moreover, HIV-C was observed in proximity of TNTs (Figure 1f), in line with recent studies, where HIV-like particles were demonstrated to spread from one DC to another through TNTs [52].

Together, these data point to the prominent role of TNTs during HIV, especially HIV-C, infection. However, at this point, whether TNTs contribute to viral spread or, on the contrary, coordinate immune cells for the containment of viral infection remains to be addressed.

### 3.2. Investigating DC and DC/CD4^+^ T-Cell Infection with HIV and HIV-C

To shed light on the role of TNT in HIV or HIV-C infection, productive infection of DCs, alone or in co-cultures with autologous naїve CD4^+^ T cells, was measured at 7 dpi. DCs only showed low-level productive infection with nonopsonized HIV, while infection with HIV-C was highly efficient (Figure 2a), in line with previous findings [17]. Despite significantly higher TNT formation in DC/T-cell co-cultures exposed to HIV-C (Figure 1a), productive infection in these co-cultures was significantly decreased compared to co-cultures exposed to HIV (Figure 2b), in agreement with recent studies that demonstrated a stronger antiviral response upon HIV-C infection [17,18,19,20,21,22,23].

### 3.3. Inhibition of TNT Formation Decreases Infection in DCs and DC/CD4^+^ Co-Cultures

To further investigate the role of TNT in HIV/HIV-C infection, DCs, alone or in co-culture with CD4^+^ T cells, were treated with cytarabine (AraC), which is a pyrimidine nucleoside analog recently demonstrated to reduce TNT formation by interfering with NFκB activation in acute myeloid leukemia (AML) [53] and in HTLV-1 cells [62]. The influence of treatment with AraC on viral infection was determined by measuring p24 concentration in supernatants at 7 dpi. A 50% infection reduction was observed in DCs, but only if they were infected with HIV-C (Figure 3a). However, when DC/CD4^+^ T-cell co-cultures were treated with AraC, HIV infection was reduced independently of its opsonization pattern (Figure 3b). Considering the importance of NFκB in immune cell maturation and development [63,64], the same experiment was repeated using a second TNT inhibitor (TNTi, NPD3064), which blocks TNT formation by interfering with the MSec protein [43].

The results from AraC and TNTi treatment were consistent with those observed using AraC. A decrease in DC infection was observed only when cells were infected with HIV-C (Figure 3c), while in DC/CD4^+^ T-cell co-cultures, productive infection was impeded using both HIV and HIV-C (Figure 3d). Together, these data demonstrate the contribution of TNTs in HIV/HIV-C spread between immune cells.

### 3.4. Antagonizing C5aR Reduces TNT Formation, Infection and Local C3 Production in HIV-C-Infected DCs and DC/CD4^+^ T-Cell Co-Cultures

Due to HIV-induced complement activation, the anaphylatoxins C3a and C5a are locally generated upon viral infection. To assess their role in TNT formation, HIV-C-infected DCs, alone and in co-cultures with CD4^+^ T cells, were treated with antagonists of C3a and C5a receptors (C3aR and C5aR). The number of TNTs displayed on DCs was counted 7 dpi, the same day when productive infection was analyzed. Lastly, internal C3 production by DCs was measured and indicated as percentages (%) of cells containing C3 of the total number of observed cells. Interestingly, antagonizing C3aR did not have a major effect on the number of TNTs displayed on the DC surface (Figure 4a), nor on the productive infection of DCs (Figure 4b). On the contrary, incubation with C5aRi led to a significant reduction in the number TNTs on DCs (Figure 4a), accompanied by significantly lower productive infection levels (Figure 4b). Quantification analysis revealed that HIV-C infection was followed by extensive induction of local C3 production in DCs (Figure 4c), with approximately 40% of cells containing C3. Complement production was clearly visible using confocal microscopy of HIV-C-infected DCs (Figure 4d).

Antagonizing C3aR and C5aR was revealed to have antithetic effects on local complement production by DCs: C3aRi resulted in an average increase of 15% of cells mobilizing intracellular C3 compared to the untreated control. On the contrary, less than 20% of C5aRi-treated cells mobilized local C3, a value that is significantly lower compared to both untreated control and C3aRi-treated DCs (Figure 4c). Similar results were obtained when HIV-C-infected DC/CD4^+^ T-cell co-cultures were treated with anaphylatoxin receptor antagonists, with only C5aRi significantly decreasing TNT formation (Figure 4a), productive infection (Figure 4b) and complement production (Figure 4c), indicating a block in *trans* infection from DCs to CD4^+^ T cells via TNTs. More detailed mechanistic insights are needed to understand these effects on a molecular level, which would have exceeded the scope of the study. Overall, these data suggest that local C3 processing has a role in enabling TNT formation and allowing enhanced interactions between cells, with C5aR having a more prominent role.

## 4. Discussion

Our study highlights the contribution of local complement activation and deposition on the HIV-1 surface for tunneling nanotube (TNT) formation in DCs. Here, we report that HIV infection increases the number of TNTs on DC surfaces, in line with previous observations in macrophages [42,43] and U87 cells [44]. Indeed, recent studies have demonstrated the ability of the accessory HIV protein Nef to hijack macrophages´ intercellular communication machinery, increasing TNT formation, via a mechanism involving Msec and MyoX that is still not completely understood [43,65,66].

Since HIV-C spontaneously accumulates in semen and during transmission and in the acute phase of infection [18,30,31,32,33], we investigated the possible involvement of complement opsonization of the virus in TNT formation. Interestingly, HIV-C mediated a significantly elevated number of TNTs on DCs compared to HIV (Figure 1a). The increased induction of TNTs and reticulation between cells was even further enhanced in DC/CD4^+^ T-cell co-cultures (Figure 1a,d), consistent with the recently illustrated helper function of CD40L on the CD4^+^ T-cell surface in the initiation of TNT formation [52]. Interestingly, immature and mature DCs did not differ in their ability to form TNTs in response to HIV/HIV-C infection, which can be explained by Nef-induced cytoskeletal arrangement, which renders immature DCs phenotypically and functionally more similar to their mature counterparts [67]. TNTs spanning from one DC to another or between DC/CD4^+^ T cells (Figure 1b,c) were observed in both HIV- and HIV-C-infected cultures, and they were particularly found between heavily infected cells (Figure 1e,f). Moreover, HIV or HIV-C were found in proximity or even inside TNTs, as previously observed in macrophages [42]. In DCs, HIV-like particles have previously been spotted traveling from one DC to another through TNTs [52].

Cell reticulation can help or harm the host depending on the context; indeed, communication mediated by TNTs can induce the immune response of target cells, which may play an important role in the activation and communication of the immune system. This intercellular transport process may, on the one hand, lead to more efficient antigen presentation and T-cell activation and is quite important for an effective cellular immune response [68,69,70]. On the other hand, many viruses, such as the influenza virus, HIV and herpes simplex virus, can elude host immunity and pharmaceutical targeting by transferring their genome to naїve cells through TNTs [42,53,62,71]. For this reason, we further investigated the effect of complement opsonization or local complement production on TNT formation and infection.

In HIV-C-infected DCs, an increase in TNT number was accompanied by a significantly higher productive infection, which, however, can be explained by the ability of HIV-C to circumvent SAMHD1 via phosphorylation at its T592 residue [17]. Since this is associated with a weakened HIV-C restriction, the resulting increase in infection may play a role in inducing TNTs through the Nef protein. However, when co-cultured with CD4^+^ T cells, the higher number of TNTs formed on DC surfaces was accompanied by a lower level of productive infection compared to co-cultures infected with nonopsonized HIV. We previously illustrated that DCs exposed to HIV-C elicit a significantly stronger type I IFN response via a CR4/CCR5/RLR (RIG-I/MDA5)/MAVS/TBK1/IRF3/NFκB signaling axis, and therefore, in co-cultures with CD4^+^ T cells, the lower level of productive infection despite higher TNT formation could be attributed to this higher type I IFN production and therefore a better antiviral response in HIV-C-infected DCs [18,19].

To shed light on the role of TNTs with respect to DC exposure to complement-opsonized HIV, we treated DCs alone or in co-cultures with CD4^+^ T cells with cytarabine (AraC), a pyrimidine nucleoside analog recently demonstrated to reduce TNT formation by interfering with NFκB activation in acute myeloid leukemia (AML) [53] and in HTLV-1 cells [62]. TNT inhibition significantly decreased infection in DCs, but only when they were infected with HIV-C but not HIV. HIV induced only low-level productive infection in DCs, which might not be as dependent on viral spread via TNTs as the elevated productive infection observed in HIV-C-exposed DC cultures. The effects mediated by AraC in HIV-C-exposed DCs or DC/T-cell co-cultures could be related to the inactivation of SAMHD1 by HIV-C or by augmented NFκB activation in HIV-C-exposed DCs [17,18,19]. Indeed, SAMHD1 was recently identified as a biomarker for the cytarabine response, with its depletion in AML cell lines associated with higher cell sensitivity to AraC cytotoxicity in therapeutic doses [72]. Although the concentration of AraC used in our experiments is far from the therapeutic dose, we cannot exclude that the observed decrease in infection is related to greater cytotoxic effects of AraC, since the enhanced infection of DCs with HIV-C was accompanied by the highly phosphorylated status of SAMHD1 T592, leading to its inactivation [17]. Additionally, treatment of DC/CD4^+^ T-cell co-cultures using AraC resulted in a decrease in infection, also using nonopsonized HIV. This reduction in HIV infection in co-cultures could be due to the downmodulation of HIV receptor expression in human T lymphoid cells upon cytarabine treatment, which ultimately also reduced cell susceptibility to HIV infection [73]. This is consistent with infection experiments that we additionally performed using CD4^+^ T cells only, where we detected significantly reduced infection using AraC and nonopsonized HIV-1 (Appendix A). As expected, HIV-C only caused low-level productive infection of CD4^+^ T cells regardless of whether they were preincubated or not with AraC due to the low-level expression of complement receptors (Appendix A).

To exclude possible cytotoxic effects of AraC on DCs due to SAMHD1 and considering the importance of NFκB in immune cell maturation and development [63,64], we used a second TNT inhibitor (TNTi), which blocks TNT formation by interfering with the MSec protein [43]. Data using TNTi confirmed the results obtained using AraC, where infection of DC cultures alone or in co-culture with T cells using HIV-C was considerably impeded or completely absent. Therefore, in particular, the antiviral effects observed in DC/CD4^+^ T-cell co-cultures using HIV-C-exposed DCs were further enhanced in the presence of TNT antagonists, independent of the mechanism of inhibition. As also observed using AraC, TNTi did not have any effect on low-level productive infection of DCs using nonopsonized HIV, while the high transmission of the virus from HIV-loaded DCs to CD4^+^ T cells was significantly impaired, indicating the strong involvement of TNTs in virus transfer. In the case of CD4^+^ T-cell infection, TNTi treatment did not show the same inhibitory effect obtained with AraC on HIV-infected-CD4^+^ T cells, which lack the MSec protein (Appendix A). This, together with the fact that HIV-infected T cells do not increase TNT production [49], led us to conclude that the inhibitory effects on HIV infection following treatment with AraC is merely due to its role in the downregulation of HIV receptors [73] and not related to TNT formation. For the same reason, the observed decrease in infection following treatment of the DC/CD4^+^ T-cell co-culture with TNTi is entirely related to the inability of DCs to reticulate.

Lastly, since we previously found that DCs exposed to HIV-C constantly produce complement-derived anaphylatoxins at mucosal sites and also locally in cells, here, we investigated the role of anaphylatoxins C5a and C3a on TNT formation and productive infection using HIV-C. C5a is a proinflammatory mediator that serves as a potent chemoattractant for iDCs and macrophages, mobilizing them to the site of viral entry during the first stages of infection [74]. We found that C5a stimulates DC reticulation, which is in line with the demonstrated association between proinflammatory stimuli and TNT formation [74]. Antagonizing C5aR led to a decline in TNTs in DC culture as well as in DC/CD4^+^ T-cell co-cultures accompanied by a reduction in viral infection and local C3 production by DCs. Blocking C3aR had opposite effects on TNT formation and viral infection, in accordance with its anti-inflammatory nature [75]. Interestingly, the same pattern was observed when C5aR and C3aR were antagonized in nonimmune epithelial cells infected with SARS-CoV-2, with the blocking of C5aR significantly diminishing TNT formation, infection and internal complement production. Importantly, upon SARS-CoV-2 infection, C3 was largely localized inside TNTs [76]. These findings, together with others, suggest that HIV has evolved a mechanism that involves complement activation to upregulate TNTs to facilitate the spread of infection. In addition to protecting the virus from the extracellular environment, utilizing TNTs to propagate infection is more energetically favorable, since the production of all viral components necessary for replication can be avoided by inserting viral genomes directly into the cytoplasm of a host cell [77].

This study provides novel insights on the mechanisms of TNT formation and utilization for effective HIV infection in terms of complement as an opsonizing agent or upon local C3 mobilization, which is vital for the development of an efficient treatment. As viruses and other pathogens have developed the ability to exploit TNTs, so too must modern medicine develop therapeutics that target TNT formation and pathogen spread through them. While we currently have the capabilities to inhibit actin polymerization, less toxic and more specific TNT inhibitors are required.

## 5. Conclusions

In this study, we report that complement opsonization of HIV-1, as well as local complement mobilization and generation of anaphylatoxin C5a, influences productive infection through TNT formation. As expected, TNT antagonists were useful in hindering *trans* infection of non- and complement-opsonized HIV-1 from DCs to T cells. Surprisingly, C5aR inhibition also significantly blocked TNT formation and intracellular C3 mobilization in DCs as well as productive infection of DCs alone and DC/T-cell co-cultures, rendering C5aR targeting a promising candidate strategy to block *trans* infection of CD4^+^ T cells, the main targets of HIV-1.

## Figures and Tables

**Figure 1 biomolecules-12-00313-f001:**
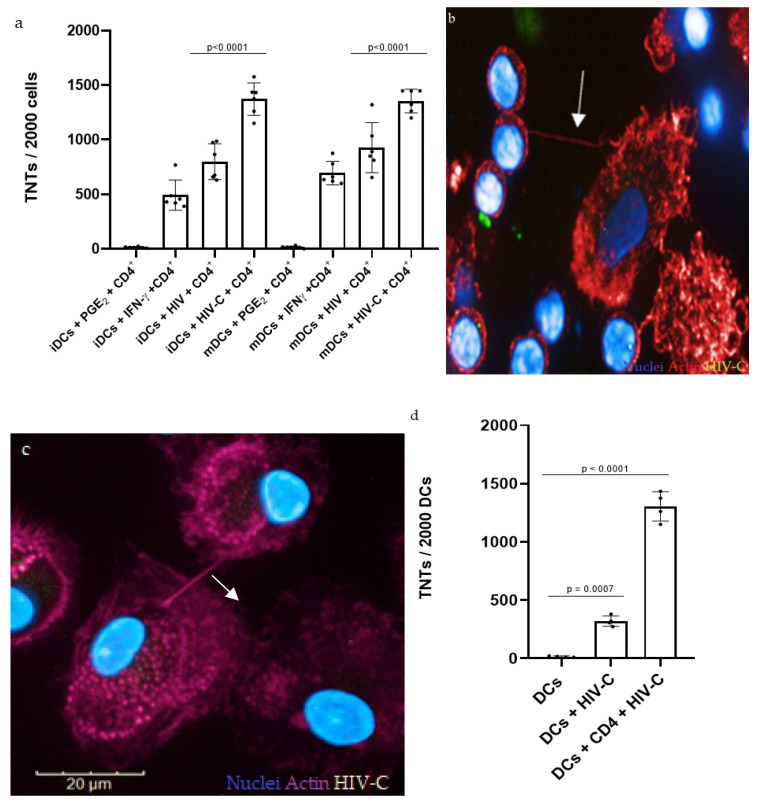
HIV-C increases TNT formation and infection in immature (i) and mature (m) DCs. (**a**) Quantification of TNTs on HIV/HIV-C-loaded i/mDCs in co-cultures with CD4^+^ T cells at 7 dpi. HIV infection induces TNT formation in both i- and mDCs, with complement opsonization of the virus significantly enhancing this tendency (*p* < 0.0001). Experiments were performed for a minimum of 3 independent replicates, and statistical significance was analyzed using GraphPad Prism software v9 [60] and ordinary one-way ANOVA. (**b**) TNT formation in GFP-HIV-C-infected DC/CD4^+^ T-cell co-culture was observed by confocal microscopy with the high-content screening Operetta system (Perkin Elmer). Nuclei were stained with Hoechst (blue), and F-actin (and therefore TNTs) was stained with phalloidin (red). HIV-C is visible in green. A TNT (highlighted with a white arrow) connects a DC and a CD4^+^ T cell. In the absence of cellular membrane staining, CD4^+^ T cells can be distinguished from DCs as round cells with a smaller diameter. Objective: 63×. The acquired image belongs to a single Z-stack, above the cells’ substratum. (**c**) TNT formation in a DC culture infected with mCherry HIV-C was observed by confocal microscopy with the high-content screening Operetta system (Perkin Elmer). Nuclei were stained with Hoechst (blue), F-actin (and therefore TNTs) was stained with phalloidin (fuchsia), and C3 was stained with FITC (green). HIV-C is visible as a yellow spot inside cells. A TNT (indicated by a white arrow) typically lies above cells’ substratum and connects two infected DCs. Objective: 63×. Z-series images were acquired at minimal Z interval and later processed together into the “Maximum projection” mode with Harmony software v 4.8 (Perkin Elmer) [58]. (**d**) TNT quantification on HIV-C-loaded DCs and DC/CD4^+^ T-cell co-cultures at 7 dpi. HIV-C-infected DCs, both alone and in a co-culture with CD4^+^ T cells, show a significantly higher number of TNTs on their surfaces. Experiments were performed for a minimum of 3 independent replicates, and statistical significance was analyzed using GraphPad Prism software v9 [60] and unpaired Student’s *t*-test. (**e**) A 3D reconstruction of a TNT connection (indicated by a white arrow) between GFP-HIV-C-infected DCs. Nuclei were stained with Hoechst (blue), and F-actin (and therefore TNTs) was stained with phalloidin (red). HIV-C is visible in green. Objective: 63×. Z-series images were acquired at minimal Z intervals and later processed together into the “Maximum projection” mode. From this image, a 3D reconstruction was obtained with the Harmony software (Perkin Elmer) v4.8 [58]. (**f**) TNT formation in GFP-HIV-C-infected DC/CD4^+^ T-cell co-culture was observed by confocal microscopy with the high-content screening Operetta system (Perkin Elmer). Nuclei were stained with Hoechst (blue), and F-actin (and therefore TNTs) was stained with phalloidin (red). HIV-C is visible in green. A TNT (indicated with a white arrow) connects two HIV-C-loaded DCs. In the absence of cellular membrane staining, CD4^+^ T cells can be distinguished from DCs as round cells with a smaller diameter. Objective: 63×. The acquired image belongs to a single Z-stack, above the cells’ substratum.

**Figure 2 biomolecules-12-00313-f002:**
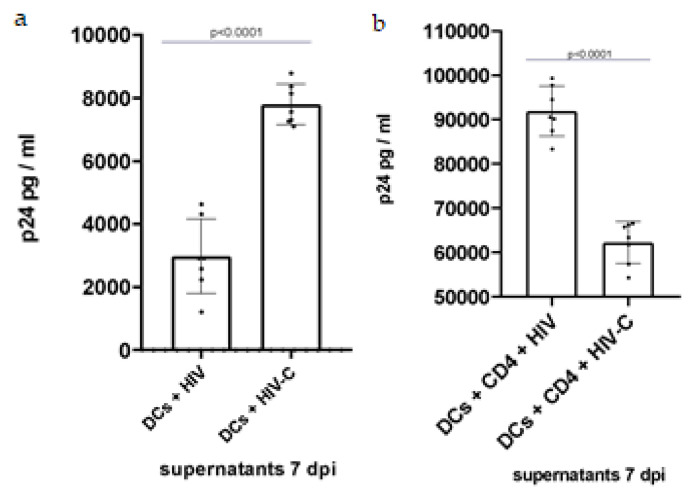
HIV/HIV-C infection of DCs and DC/CD4^+^ T-cell co-cultures. (**a**) Quantification of infection in 7 dpi supernatants from HIV/HIV-C-infected DCs. DCs incubated with HIV show significantly decreased, moderate productive infection compared to HIV-C (*p* < 0.0001). Experiments were performed for a minimum of 3 independent replicates, and statistical significance was analyzed using GraphPad Prism software v9 [60] and unpaired Student’s *t*-test. Abbreviation: dpi: days postinfection. (**b**) Quantification of infection in 7 dpi supernatants from HIV/HIV-C-infected DC/CD4^+^ T-cell co-culture. Infection with HIV-C is decreased and moderate compared to the control infection with HIV (*p* < 0.0001). Experiments were performed for a minimum of 3 independent replicates, and statistical significance was analyzed using GraphPad Prism software v9 [60] and unpaired Student’s *t*-test.

**Figure 3 biomolecules-12-00313-f003:**
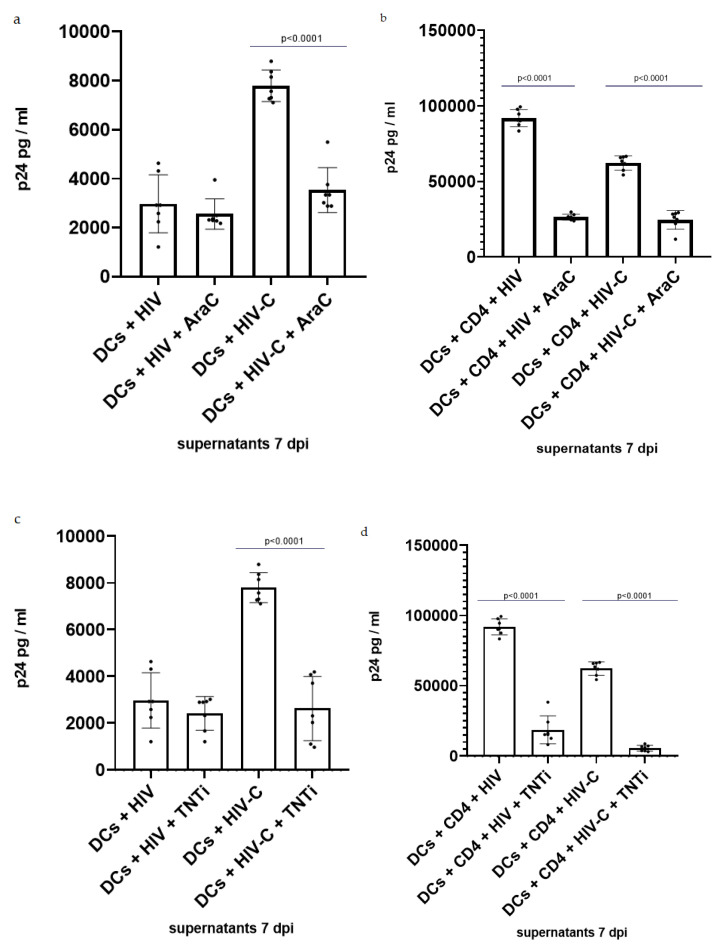
TNT inhibition decreases infection in HIV-C-loaded DCs and in DC/CD4^+^ T-cell co-culture independently of virus opsonization pattern. (**a**) Quantification of infection in 7 dpi supernatants from HIV/HIV-C-infected DCs with/without 24 h treatment with 1 µM AraC. HIV-C-infected DCs treated with AraC show significantly decreased, moderate productive infection compared to their untreated counterparts (*p* < 0.0001). Experiments were performed for a minimum of 3 independent replicates, and statistical significance was analyzed using GraphPad Prism software v9 [60] and ordinary one-way ANOVA. (**b**) Quantification of infection in 7 dpi supernatants from HIV/HIV-C-infected DC/CD4^+^ T-cell co-cultures with/without 24 h treatment with 1 µM AraC. The inhibition of infection is observed to be independent of HIV opsonization (*p* < 0.0001). Experiments were performed for a minimum of 3 independent replicates, and statistical significance was analyzed using GraphPad Prism software v9 [60] and ordinary one-way ANOVA. (**c**) Quantification of infection in 7 dpi supernatants from HIV/HIV-C-infected DCs with/without 24 h treatment with 20 µM TNTi. HIV-C-infected DCs treated with TNTi show significantly decreased, moderate productive infection compared to their non-treated counterparts (*p* < 0.0001). Experiments were performed for a minimum of 3 independent replicates, and statistical significance was analyzed using GraphPad Prism software v9 [60] and ordinary one-way ANOVA. (**d**) Quantification of infection in 7 dpi supernatants from HIV/HIV-C-infected DC/CD4^+^ T-cell co-cultures with/without 24 h treatment with 20 µM TNTi. The inhibition of infection is observed to be independent of HIV opsonization (*p* < 0.0001; *p* < 0.001). Experiments were performed for a minimum of 3 independent replicates, and statistical significance was analyzed using GraphPad Prism software v9 [60] and ordinary one-way ANOVA.

**Figure 4 biomolecules-12-00313-f004:**
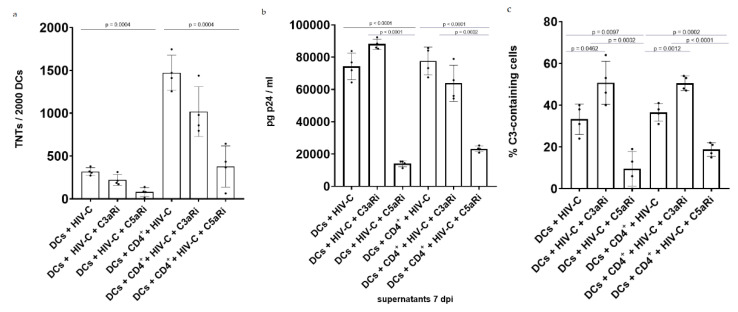
Treatment of HIV-C-infected DCs and DC/CD4^+^ T-cell co-cultures with antagonists of anaphylatoxin receptors C3aRI and C5aRI/II. (**a**) Quantification of TNTs on HIV-C-loaded DCs and DC/CD4^+^ T-cell co-cultures at 7 dpi with/without treatment with C3aRi and C5aRi. Antagonizing C3aR does not show any effect on TNT formation, while C5aRi-treated DCs display a significant reduction in TNT on their surfaces (*p* = 0.0004; *p* = 0.0004). Experiments were performed for a minimum of 2 independent replicates, and statistical significance was analyzed using GraphPad Prism software v9 [60] and ordinary one-way ANOVA. (**b**) Quantification of productive infection in HIV-C-infected DCs and DC/CD4^+^ T-cell co-cultures with/without treatment with C3aRi and C5aRi. C3aRi-treated DCs show levels of productive infection with HIV-C equal to the untreated control, while C5aRi-treated cells show decreased infection with HIV-C (*p* < 0.0001; *p* < 0.0001). Experiments were performed for a minimum of 2 independent replicates, and statistical significance was analyzed using GraphPad Prism software v9 [60] and ordinary one-way ANOVA. (**c**) DC complement production was measured as a percentage of C3-containing cells of all observed cells. C3aRi treatment significantly increased the percentage of cells containing C3 (*p* = 0.0462; *p* = 0.0012). On the contrary, C5aRi decreased the percentage of cells producing C3 compared to both the untreated control (*p* = 0.0097; *p* = 0.0002) and C3aRi-treated DCs (*p* = 0.0002; *p* < 0.0001). (**d**) A 3D reconstruction of a culture of DCs infected with mCherry HIV-C (in yellow). C3 is stained with FITC (green), F-actin (therefore TNTs) is stained with phalloidin (fucshia), and nuclei are stained with Hoechst (blue). A TNT between two productively HIV-C-infected DCs is shown with a white arrow. Objective: 63×. Z-series images were acquired at minimal Z intervals and later processed together into the “Maximum projection” mode. From this image, a 3D reconstruction was obtained with the Harmony software v4.8 [58] (Perkin Elmer).

## Data Availability

Not applicable.

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
