# Peer review of "HIV-1 Trans Infection via TNTs Is Impeded by Targeting C5aR"

_biomolecules, 2022, doi:10.3390/biom12020313_

Round 1
Reviewer 1 Report
Summary of findings
The authors start by showing that HIV can drive the formation of TNTs in MDDC/CD4 T cell cocultures similar to previously shown in macrophages and cell lines. The effect is observed using both immature and mature MDDCs and the effect is more pronounced when using opsinized HIV (HIV-C). To a lesser extent TNTs between MDDCs (in the absence of CD4 T cells) were also observed. Importantly, the effect was more marked in MDDCs with more virions present and HIV-C could be observed in proximity to and inside TNTs (better images are needed). The authors then show that HIV-C is more efficient at productively infecting MDDCs than HIV but infection was lower when the DCs were cocultured with CD4 T cells. They then show that inhibiting TNT formation using cytarabine reduced infection of MDDCs with HIV-C (but not HIV), but in DC co-cultures cytarabine reduced infection when using HIV and HIV-C. Similar results were obtaining using a second TNT inhibitor (TNTi, NPD3064). Finally, the authors show that antagonizing C5aR reduces TNT formation, infection and local C3 production in HIV-C infected DCs and DC/CD4+ T cells co-cultures.
Overall, this is a nice story that is well designed and implemented and logical series of experiments. I believe this will be of interest to the readers of this journal.
My comments are:
The abstract is long winded and could be shortened to be more concise.
Introduction
The authors state “Upon contact with the invading pathogen, DCs mature into potent antigen-presenting cells (APCs) and migrate to lymph nodes, where they present the acquired antigen to naive CD4+ T cells”. Recent studies have shown that transfer of HIV from CD4 T cells occurs as early as 2 hours after exposure (e.g. PMID: 33846309 & 31227717). This early transfer likely occurs in the mucosa not the lymph nodes. Thus, the statement should be reworded to something along the lines of: “Upon contact with the invading pathogen, DCs mature into potent antigen-presenting cells (APCs) and present the acquired antigen to naive CD4+ T cells either in mucosal tissue or after migrate to lymph nodes:
The authors state: “Moreover, DCs exposed to HIV-C display more profound maturation and co-stimulatory capacity [17] as well as an increased production of type I interferon [18-19] compared to DCs exposed to non-opsonized HIV-1”. In fact DC exposed to HIV do not produced any interferon at all (PMID: 25855743 & 21411754). The sentence should be reworded to something along the lines of “Moreover, DCs exposed to HIV-C display more profound maturation and co-stimulatory capacity [17] as well as production of type I interferons [18-19] compared to DCs exposed to non-opsonized HIV-1 which do not produce Type I or III interferons”.
Minor points:
- The manuscript by the Turville laboratory should be briefly mentioned in the introduction or discussion as it is relevant to this study:
https://pubmed.ncbi.nlm.nih.gov/22685410/
https://pubmed.ncbi.nlm.nih.gov/28321960/
- Viral titres were and infectivity assays were determined by p24 ELISA which detects the amount of 24 protein present (i.e. the virions released from cells) but does not directly measure infectivity. Is there a reason why DC infectivity was not measured using flow cytometry which directly measures HIV infectivity? It can also be used to differentiate between HIV uptake (observed as a smear in p24 fluorescence) vs infection (observed as a discrete and separate population of HIV infected cells).
- Observing TNTs on Day 7 is a long time after infection and co-culture. Can the authors comment on the physiological relevance of this? Have they previously attempted to visualise this at earlier time points? DCs and LCs can transfer HIV to CD4 T cells as early as 2 hours post infection (e.g. PMID 33846309, 31227717, 25070850, 20571487).
- Many of the statements made in relation to Figure 1 are based on qualitative observations and the conclusions/inferences should be more cautionary, or the results quantified. For examples, the authors state “TNTs especially formed between heavily infected cells (Figure 1 (d))”. This is a very qualitative observations based on 1 image. Representative images should be shown from cells containing both high and low levels of virus partials to substantiate this claim. It would be better still if a quantitative analysis could be carried out. Similarly, they state “HIV-C was often found in proximity or even inside TNTs (Figure 1 (e)). Again, this is based on one image and it isn’t clear even if HIV is in the TNT. Better multiple images are needed
- Given immature DCs are functionally different to mature DCs, can the authors hypothesise as to why do not differ in their ability to form TNTs with CD4 T cells in response to HIV and HIV-C?
- Co-culture with CD4 T cells showed increased levels of TNT formation and productive infection. The authors suggest this may be due to the recent findings of CD40L on CD4 T cells initiating TNT formation. Could this be investigated through a CD40L blocking assay?
- In the discussion the authors state “This, together with the fact that HIV-infected T cells do not increase TNT production [48], let us conclude that the inhibitory effects on HIV infection following treatment with AraC is merely due to its role in downregulation of HIV-receptors and not related to TNT formation”. Is it possible to look at HIV receptors on these cells pre and post treatment and identify this hypothesised downregulation?
- In the discussion the authors state “Antagonizing C5aR led to a decline in TNTs in DCs culture as well as in DC/CD4+ T cells co-cultures accompanied by reduction in viral infection and local C3 production by DCs”. The data presented on DC/CD4+ T cells with C5aR treatment only included productive infection (pg p24/ml), the data for TNT formation or C3 production were not presented. This data would be of great interest to be included in figure 4a/b/c.
Minor points
- Some of the in text references to figures are incorrect as below:
- Bottom of page 8, Figure 3a and Figure 3b should read Figure 3c and Figure 3d respectively
- Middle of page 10, Figure 6a and Figure 6b should read Figure 4a and Figure 4b.
- Figure 1f seems to be in a weird placement of order. In text it is referenced prior to figure 1c and 1d
- Would be helpful having legends for microscopy images defining the colours
- Figure 4d it is a bit hard to determine what is HIV (yellow) alongside the fuschia and green, which when overlapping creates a yellow/white
Author Response
We want to thank the Reviewer for taking the time to review this manuscript and we feel that the manuscript improved by including the Reviewer´s comments. We addressed the concerns that have been raised in the review as follows:
- The abstract is long winded and could be shortened to be more concise.
We shortened the abstract to less than 300 words and made it more concise.
Introduction
The authors state “Upon contact with the invading pathogen, DCs mature into potent antigen-presenting cells (APCs) and migrate to lymph nodes, where they present the acquired antigen to naive CD4+ T cells”. Recent studies have shown that transfer of HIV from CD4 T cells occurs as early as 2 hours after exposure (e.g. PMID: 33846309 & 31227717). This early transfer likely occurs in the mucosa not the lymph nodes. Thus, the statement should be reworded to something along the lines of: “Upon contact with the invading pathogen, DCs mature into potent antigen-presenting cells (APCs) and present the acquired antigen to naive CD4+ T cells either in mucosal tissue or after migrate to lymph nodes:
The authors state: “Moreover, DCs exposed to HIV-C display more profound maturation and co-stimulatory capacity [17] as well as an increased production of type I interferon [18-19] compared to DCs exposed to non-opsonized HIV-1”. In fact DC exposed to HIV do not produced any interferon at all (PMID: 25855743 & 21411754). The sentence should be reworded to something along the lines of “Moreover, DCs exposed to HIV-C display more profound maturation and co-stimulatory capacity [17] as well as production of type I interferons [18-19] compared to DCs exposed to non-opsonized HIV-1 which do not produce Type I or III interferons”.
Both the statements have been corrected as suggested and the reference studies have been cited.
-
The manuscript by the Turville laboratory should be briefly mentioned in the introduction or discussion as it is relevant to this study:
https://pubmed.ncbi.nlm.nih.gov/22685410/
https://pubmed.ncbi.nlm.nih.gov/28321960/
Turville et al manuscripts have been cited in the introduction as follows:
“Actin remodeling influence on HIV infection has already been demonstrated in CD4+ T cells, where dynamin inhibition has been linked to an impaired cells’ infection [50; https://pubmed.ncbi.nlm.nih.gov/22685410/].
In DCs, budding HIV and not mature HIV particles have been observed moving inside long actin-rich filopodia, not consistent with TNT structure, which converge to form viral synapses with CD4+ T cells [51; https://pubmed.ncbi.nlm.nih.gov/28321960/]”.
-
Viral titres were and infectivity assays were determined by p24 ELISA which detects the amount of 24 protein present (i.e. the virions released from cells) but does not directly measure infectivity. Is there a reason why DC infectivity was not measured using flow cytometry which directly measures HIV infectivity? It can also be used to differentiate between HIV uptake (observed as a smear in p24 fluorescence) vs infection (observed as a discrete and separate population of HIV infected cells).
In this work, p24 ELISA results have been considered as a measure of productive infection by HIV/-C and we wanted to detect released virus within the co-cultures. To avoid any misinterpretation, cells were thoroughly washed to efficiently detect only productive infection by HIV/-C and not cell-bound viruses on several days post infection. The suggestion to use FACS to detect HIV infection, so to distinguish it from viral uptake, will be surely considered in future experiments.
-
Observing TNTs on Day 7 is a long time after infection and co-culture. Can the authors comment on the physiological relevance of this? Have they previously attempted to visualise this at earlier time points? DCs and LCs can transfer HIV to CD4 T cells as early as 2 hours post infection (e.g. PMID 33846309, 31227717, 25070850, 20571487).
TNTs were quantified 7 days after infection based on the results of preliminary experiments. These results have now been added in the supplementary files (Figure S1). Here, differentially stimulated immature and mature DCs were infected with HIV or HIV-C and TNTs were counted at different time points: 24 h, 48 h, 4 d and 7 d. Since the quantity of TNTs was at the highest at 7 dpi compared to the other tested time points, we decided to measure productive infection and complement production at 7 dpi.
-
Many of the statements made in relation to Figure 1 are based on qualitative observations and the conclusions/inferences should be more cautionary, or the results quantified. For examples, the authors state “TNTs especially formed between heavily infected cells (Figure 1 (d))”. This is a very qualitative observations based on 1 image. Representative images should be shown from cells containing both high and low levels of virus partials to substantiate this claim. It would be better still if a quantitative analysis could be carried out. Similarly, they state “HIV-C was often found in proximity or even inside TNTs (Figure 1 (e)). Again, this is based on one image, and it isn’t clear even if HIV is in the TNT. Better multiple images are needed
Cautionary words have been added to the statements, highlighting that they derive purely from qualitative observations of images. The statement “TNTs especially formed between heavily infected cells” has been corrected in “From qualitative observations, TNTs seem to especially form between heavily infected cells” and “HIV-C was often found in proximity or even inside TNTs“ has been corrected to “Moreover, HIV-C was observed in proximity of TNTs”. Regarding the possibility to quantify these results, a high-throughput quantitative analysis will soon be run in the lab.
-
Given immature DCs are functionally different to mature DCs, can the authors hypothesise as to why do not differ in their ability to form TNTs with CD4 T cells in response to HIV and HIV-C?
The same ability of immature and mature DCs to form TNTs with CD4+ T cells in response to HIV and HIV-C could be explained with the ability of the viral protein Nef to induce a series of phenotypical, morphological, and functional changes that renders immature DCs very similar to mature DCs. In addition, Nef has been demonstrated to cause profound cytoskeletal rearrangements, which also include a higher number of F-actin-rich cell surface structures. This explanation and the reference have been properly cited in the discussion as follows: “Interestingly, immature and mature DCs did not differ in their ability to form TNTs in response to HIV/-C infection, which can be explained by Nef-induced cytoskeletal arrangement, that renders immature DCs phenotypically and functionally more similar to their mature counterpart [64]” (https://doi.org/10.1096/fj.03-0272com).
-
Co-culture with CD4 T cells showed increased levels of TNT formation and productive infection. The authors suggest this may be due to the recent findings of CD40L on CD4 T cells initiating TNT formation. Could this be investigated through a CD40L blocking assay?
This is a very good suggestion and will surely be taken in consideration, but it would have exceeded the scope of the study. Mechanistic insights into TNT formation in DCs exposed to differentially opsonized HIV are now ongoing in our lab.
-
In the discussion the authors state “This, together with the fact that HIV-infected T cells do not increase TNT production [48], let us conclude that the inhibitory effects on HIV infection following treatment with AraC is merely due to its role in downregulation of HIV-receptors and not related to TNT formation”. Is it possible to look at HIV receptors on these cells pre and post treatment and identify this hypothesised downregulation?
In this study, the downregulation of HIV-receptors has not been tested, but it has been demonstrated by Gröschel et al., whose work (doi: 10.1081/NCN-100002571, reference number 70 in the manuscript) has been cited in the “Discussion” as follows: “This reduction of HIV-infection in co-cultures could be due to down-modulating HIV-receptor expression in human T lymphoid cells upon cytarabine treatment, which ultimately also reduced cells susceptibility to HIV infection [70]” and “This, together with the fact that HIV-infected T cells do not increase TNT production [49], let us conclude that the inhibitory effects on HIV infection following treatment with AraC is merely due to its role in downregulation of HIV-receptors [70] and not related to TNT formation”.
-
In the discussion the authors state “Antagonizing C5aR led to a decline in TNTs in DCs culture as well as in DC/CD4+ T cells co-cultures accompanied by reduction in viral infection and local C3 production by DCs”. The data presented on DC/CD4+ T cells with C5aR treatment only included productive infection (pg p24/ml), the data for TNT formation or C3 production were not presented. This data would be of great interest to be included in figure 4a/b/c.
The data for TNT formation and C3 production for DC/CD4+ T cell co-cultures with C3aR and C5aR treatment have been added in figure 4a, b and c.
Minor points
-
Some of the in-text references to figures are incorrect as below:
-
Bottom of page 8, Figure 3a and Figure 3b should read Figure 3c and Figure 3d respectively
-
Middle of page 10, Figure 6a and Figure 6b should read Figure 4a and Figure 4b.
The in-text references to figures have been corrected, thanks for the notice.
-
Figure 1f seems to be in a weird placement of order. In text it is referenced prior to figure 1c and 1d
Thanks - Figures 1 have been ordered so that they are cited in an alphabetical order in the “Results” section.
-
Would be helpful having legends for microscopy images defining the colours
Legends for microscopic images defining the colours have been added close to the pictures.
-
Figure 4d it is a bit hard to determine what is HIV (yellow) alongside the fuchsia and green, which when overlapping creates a yellow/white.
We are sorry for this colour combination, but unfortunately, due to a problem with the software dongle to process the figure, at the moment, we don’t have the possibility to change the colour of figure 4d. Maybe we can hand in this figure to the MDPI office at a little later time point, when we got it repaired, that they can exchange the pics accordingly.

Reviewer 2 Report
The manuscript HIV-1 trans infection via TNTs is impeded by targeting C5aR deals with the TNT formation displayed on the surface of DCs or DC/CD4+ T cell co-cultures incubated with non- or complement-opsonized HIV-1 and the role of TNTs or locally produced complement in the infection process using either two different TNT or anaphylatoxin receptor antagonists. Congratulations on this magnificent work: the manuscript is well written, well structured, and potentially interesting in the field of prevention of HIV transmission. I just have a few minor suggestions:
Abstract: Please, make it a little shorter, +300 words is too much for an abstract.
Materials and methods: Please, indicate the meaning of FACS (Fluorescence-activated cell sorting) the first time you mention it. In the Virus Capture Assay, please, mention the authors who first described the methodology (i.e., “as described by Wilflingseder et al” or “as previously described by our team”).
Results: You already described the meaning of dpi (days post-infection) the first time you mention it, so do not repeat it in the Figure 2 caption.
Author Response
We want to thank the Reviewer or taking the time to review this manuscript.
We addressed all the concerns that has been raised in the review as follows:
Abstract: Please, make it a little shorter, +300 words is too much for an abstract.
The abstract has been now shortened to less than 300 words to make it more concise.
Materials and methods: Please, indicate the meaning of FACS (Fluorescence-activated cell sorting) the first time you mention it.
The meaning of FACS has been added the first time it is cited in “Materials and Methods”.
In the Virus Capture Assay, please, mention the authors who first described the methodology (i.e., “as described by Wilflingseder et al” or “as previously described by our team”).
The phrase “as previously described by our team” was added in the “Virus Capture Assay” section.
Results: You already described the meaning of dpi (days post-infection) the first time you mention it, so do not repeat it in the Figure 2 caption.
Repetitions in figure legends have been removed.